REGISTERED REPORT

# Registered report: Melanoma genome sequencing reveals frequent *PREX2* mutations

**Denise Chroscinski[1], Darryl Sampey[2], Alex Hewitt[3], Reproducibility Project: Cancer Biology\*†**

[1]Noble Life Sciences, Gaithersburg, United States; [2]BioFactura, Frederick, United States; [3]Department of Clinical Genetics, University of Melbourne, Melbourne, Australia

**Abstract** The Reproducibility Project: Cancer Biology seeks to address growing concerns about reproducibility in scientific research by conducting replications of 50 papers in the field of cancer biology published between 2010 and 2012. This Registered Report describes the proposed replication plan of key experiments from 'Melanoma genome sequencing reveals frequent *PREX2* mutations' by Berger and colleagues, published in *Nature* in 2012 (*Berger et al., 2012*). The key experiments that will be replicated are those reported in Figure 3B and Supplementary Figure S6. In these experiments, Berger and colleagues show that somatic *PREX2* mutations identified through whole-genome sequencing of human melanoma can contribute to enhanced lethality of tumor xenografts in nude mice (Figure 3B, S6B, and S6C; *Berger et al., 2012*). The Reproducibility Project: Cancer Biology is a collaboration between the Center for Open Science and Science Exchange, and the results of the replications will be published by *eLife*.

**\*For correspondence:** joelle@ scienceexchange.com

**Group author details**
†Reproducibility Project: Cancer Biology
See page 14

**Reviewing editor**: Roger Davis, University of Massachusetts Medical School, United States

## Introduction

Melanoma is a highly aggressive tumor with poor prognosis in the metastatic stage. Based on their association with UV-induced DNA damage, melanomas are often hypermutated and considerable efforts have been made to sequence such tumors in order to better understand their molecular basis. Many well-known oncogenes are frequently involved in melanoma pathogenesis, including *BRAF* and *NRAS*, and significant work has been done to develop targeted kinase inhibitors against the protein products of these genes (*Kunz, 2014*). However, even with treatment, melanoma has an extremely high rate of recurrence; thus, there is great interest in identifying novel candidate genes that promote oncogenesis in melanoma, thereby providing additional therapeutic targets.

One such candidate is Phosphatidylinositol-3,4,5-trisphosphate RAC Exchanger 2 (PREX2), a 183-kDa protein known to inhibit PTEN phosphatase activity, stimulate PI3K signaling, and suspected to regulate the small GTPase RAC1 (*Fine et al., 2009*; *Cerami et al., 2012*). Using whole-genome sequencing of 25 metastatic tumors, Berger and colleagues identified *PREX2* as being a highly mutated gene in melanoma. Apart from observing a large subset of *BRAF* and *NRAS* mutations, the authors found *PREX2* to have a mutation frequency of approximately 14%, with 13 detected non-synonymous point mutations, including four nonsense truncation mutations (*Berger et al., 2012*). In order to demonstrate the biological relevance of specific *PREX2* mutations, the authors created transformed melanocyte cell lines that stably expressed various mutated and truncated forms of PREX2. By using these cell lines to create tumor xenografts in nude mice, the authors showed that ectopic expression of mutant PREX2 accelerated tumor formation.

Berger and colleagues chose to analyze six representative *PREX2* mutations derived from their whole-genome sequencing screen. These variants included three truncation variants and three non-synonymous point mutations predicted to carry functional impact. These mutant *PREX2* constructs were packaged into lentiviruses and transduced into TERT-immortalized human melanocytes engineered to express $NRAS^{G12D}$. Ectopic expression of various mutant PREX2 isoforms was confirmed by Western blot (Figure 6A). These experiments will be replicated in Protocols 1 and 2. Berger and colleagues next transplanted the melanocytic lines into immunodeficient mice alongside control melanocytes expressing either wild-type PREX2 or GFP (green fluorescent protein). They found that overexpression of all three truncated variants, as well as the point mutation G844D, significantly accelerated tumor growth in vivo, thus affirming the biological relevance of their genomic data (Figure 3B, S6B, and S6C). These key experiments, which support the hypothesis that mutant *PREX2* promotes oncogenesis in melanoma, will be replicated in Protocol 3.

There is some debate over which mutations observed in various melanoma samples are biologically relevant, including *PREX2*. Potentially, mutational heterogeneity across tumor samples may contribute to false-positive findings (*Lawrence et al., 2013*). Various genome-wide screens have yielded conflicting results about which genes are frequently mutated in melanoma. Recently, mutated *PREX2* was identified in both the primary tumor and in metastatic tumor tissue from a genomic analysis of a single melanoma patient (*Turajlic et al., 2012*). However, five studies failed to identify *PREX2* in their genome-wide melanoma screens, including a meta-analysis study that analyzed hundreds of published datasets (*Hodis et al., 2012*; *Krauthammer et al., 2012*; *Ni et al., 2013*; *Marzese et al., 2014*; *Xia et al., 2014*). To date, there have been no replication attempts assessing the biological significance of PREX2 mutant isoforms in melanoma.

## Materials and methods

Unless otherwise noted, all protocol information was derived from the original paper, references from the original paper, or information obtained directly from the authors.

### Protocol 1: generation of $NRAS^{G12D}$ melanocyte cells expressing various mutated forms of PREX2

This protocol describes the generation of pMEL/hTERT/CDK4(R24C)/p53DD/$NRAS^{G12D}$ ($NRAS^{G12D}$) melanocytes that stably express various mutated forms of PREX2. This protocol details the production of lentivirus for each mutated PREX2 isoform, as well as the viral transduction of melanocytes, and selection for stable-expressing lines using antibiotic resistance.

### Sampling

- Outline of experimental endpoints:

  1. At the end of this protocol, we will have generated $NRAS^{G12D}$ melanocytes overexpressing the following protein products:

  - GFP vector (*control*)
  - WT PREX2 (*control*)
  - PREX2 Q1430* (*Truncation mutation*)
  - PREX2 G844D (*Substitution mutation*)

### Materials and reagents

| Reagent | Type | Manufacturer | Catalog # | Comments |
|---|---|---|---|---|
| GenElute Endotoxin-free Plasmid Maxiprep Kit | Reagent | Sigma | PLEX15-1KT | This kit replaces the Qiagen Endo-free Maxiprep kit used by the original authors |
| pMD2-Gag/Pol | Viral packaging vector | N/A | N/A | Reagent being provided by original authors |

*Table 1. Continued on next page*

*Table 1. Continued*

| Reagent | Type | Manufacturer | Catalog # | Comments |
|---|---|---|---|---|
| pMD2 VSVG | Viral packaging vector | N/A | N/A | Reagent being provided by original authors |
| RSV REV | Viral packaging vector | N/A | N/A | Reagent being provided by original authors |
| GFP | Expression construct | N/A | N/A | Reagent being provided by original authors |
| Wild-type PREX2 | Expression construct | N/A | N/A | Reagent being provided by original authors |
| PREX2 Q1430* | Expression construct | N/A | N/A | Reagent being provided by original authors |
| PREX2 G844D | Expression construct | N/A | N/A | Reagent being provided by original authors |
| HEK293T cells | Cell line | ATCC | CRL-3216 | Replaces original cells from Life Technologies |
| NRAS$^{G12D}$ melanocytes | Cell line | N/A | N/A | Reagent being provided by original authors |
| Sequencing primers | Oligos | Sequences provided by original authors; specific brand information will be left up to the discretion of the replicating lab and recorded later | | |
| Sequencing reagents | Reagent | Specific brand information will be left up to the discretion of the replicating lab and recorded later | | |
| 10 cm tissue culture dishes (plastic) | Labware | Corning (Sigma-Aldrich) | CLS430167 | Original brand not specified |
| 10 cm tissue culture dishes (glass) | Labware | Corning (Sigma-Aldrich) | CLS70165101 | Additional reagent not used in original study |
| Fetal bovine serum (FBS) | Cell culture reagent | Sigma-Aldrich | F0392 | Replaces Invitrogen cat. no. 26400-036 used in original study |
| Dulbecco's Modified Eagle's Medium (DMEM) – high glucose | Cell culture reagent | Sigma-Aldrich | D6429 | Replaces Invitrogen cat. no. 11995-065 used in original study |
| Lipofectamine 2000 | Transfection reagent | Life Technologies | 52887 | |
| OptiMEM-1 reduced serum medium | Cell culture reagent | Life Technologies | 31985-070 | |
| Ham's F10 medium | Cell culture reagent | Sigma-Aldrich | N6908 | Replaces Invitrogen cat. no. 11550-043 used in original study |
| Fetal bovine serum (FBS); heat inactivated | Cell culture reagent | Sigma-Aldrich | F4135 | Replaces Invitrogen cat. no. 10082-147 used in original study |
| Penicillin–Streptomycin solution (100x) stabilized | Cell culture reagent | Sigma-Aldrich | P4333 | Replaces Invitrogen cat. no. 15140-122 used in original study |
| 6 cm tissue culture dishes | Labware | Corning (Sigma-Aldrich) | CLS430166 | Original brand not specified |
| Hexadimethrine bromide (Polybrene) | Cell culture reagent | Sigma-Aldrich | 107689 | Original brand not specified |
| Blasticidin S, hydrochloride | Antibiotic | EMD-Millipore | 203350 | Original brand not specified |

*Table 1. Continued on next page*

*Table 1. Continued*

| Reagent | Type | Manufacturer | Catalog # | Comments |
|---------|------|--------------|-----------|----------|
| TRI reagent | Reagent | Sigma-Aldrich | T9424 | Additional reagent not used in original study |
| SuperScript III First-Strand Synthesis System | cDNA synthesis | Life Technologies | 18080-051 | Additional reagent not used in original study |
| Nuclease-Free Water (not DEPC treated) | Reagent | Life Technologies | AM9930 | Additional reagent not used in original study |
| RNase AWAY (spray) | Reagent | Fisher | 21-402-178 | Additional reagent not used in original study |

## Procedure

Note: all cell lines will be sent for STR profiling and mycoplasma testing.

1. Grow and prepare endotoxin-free plasmid constructs according to the manufacturer's protocol for the GenElute Endotoxin-free Plasmid Maxiprep Kit.

   A. Viral packaging vectors:

   i. pMD2-Gag/Pol (~25 µg DNA needed for production of 4 viruses)
   ii. pMD2 VSVG (~15 µg DNA needed for production of 4 viruses)
   iii. RSV REV (~17 µg DNA needed for production of 4 viruses)

   B. PREX2 expression vectors:

   i. GFP vector (~15 µg DNA needed for virus production)
   ii. WT PREX2 (~15 µg DNA needed for virus production)
   iii. PREX2 Q1430* (~15 µg DNA needed for virus production)
   iv. PREX2 G844D (~15 µg DNA needed for virus production)

2. Sequence *PREX2* plasmids to confirm identity and run on gel to confirm vector integrity. Use the following sequencing primers:

   A. CMV forward: CGCAAATGGGCGGTAGGCGTG
   B. prex2a-1 forward: ACTGAAATGCTAATGTGTGG
   C. prex2a-2 forward: CCTTTTTACTCCAGTGATAAGAGAT
   D. prex2a-3 forward: AGTACAGGCGGCCAACGAAG
   E. prex2a-4 forward: ATCACAACCATGGCGGCCCCTT
   F. prex2a-5 forward: GTAGGCTACTCCTGGCTCTT
   G. prex2a-6 forward: AGCTGCCTGTGCAAACACAG
   H. prex2a-7 reverse: GACTTCCTTCTGCTTGATAT
   I. prex2a-8 reverse: TGCTGGTGAAGGAGGCGATG
   J. prex2a-9 reverse: AGAGAATTTAGGCTGGTACA
   K. prex2a-10 reverse: ATCCCTTTTCTACCAACTTT
   L. prex2a-11 reverse: CTTGCTCCATTCCTAATTTT
   M. prex2a-12 reverse: CCTTCTCATGGTTACTACAATATTC
   N. V5 reverse: ACCGAGGAGAGGGTTAGGGAT

3. Using the same primers as above, sequence the endogenous *PREX2* gene from cDNA derived from untransfected pMEL/hTERT/CDK4(R24C)/p53DD/NRAS[G12D] melanocytes.

   A. Melanocytes should be maintained in Ham's F10 medium supplemented with 10% heat inactivated FBS and 1% penicillin/streptomycin at 37°C with 5% $CO_2$.
   B. Isolate total RNA using TRI reagent, and generate cDNA as described in the manufacturer's protocol for SuperScript III cDNA synthesis kit, using OligoDT primers to enrich for mRNA.
   C. Use gene-specific primers to sequence the length of the *PREX2* gene to determine endogenous mutational status.

4.  On Day 1 of viral production, plate $6 \times 10^6$ HEK293T cells in a 10 cm plate. Plate one 10-cm plate for each virus you wish to package (total of 4 plates needed).

    A. HEK293T cells should be maintained in DMEM supplemented with 10% FBS at 37°C with 5% $CO_2$.
    B. Note: high titer lentivirus is best packaged in early passage, healthy 293T cells. Avoid continuous growth to/from confluence. Routinely split 293T when culture approaches 80% confluence.

5.  On Day 2, create the transfection master mix: (Tube #1)

    A. Create a master mix (for the number of transfections being conducted) of Lipofectamine and OptiMEM.

        i. Each transfection will require 30 µl of Lipofectamine diluted in 720 µl of OptiMEM. Allow mixture to incubate for 5 min at RT.

6.  For each virus, assemble DNA, packaging vectors, and OptiMEM in a 1.5 ml centrifuge tube (Tube #2)

    A. Plasmid DNA = 10.0 µg
    B. Packaging vector

        i. pMD2 Gag/Pol = 5.0 µg
        ii. pRESREV = 2.5 µg
        iii. pMD2 VSVG = 3.0 µg

    C. Bring volume to 750 µl with OptiMEM.

7.  Combine Tube #1 (Lipofectamine/OptiMEM) with Tube #2 (DNA/packaging vector/OptiMEM). After combining, mix by pipetting and allow the mixture to incubate for 20 min at RT.

    A. While incubating, 'gently' aspirate growth medium from HEK293T cells and pipette 8 ml of OptiMEM to each plate.
    B. Add 1.5 ml of transfection mixture to the plate (pipetting directly into the media) and place into the 37°C incubator.
    C. Allow minimum 6–8 hr for transfection. After transfection completion, remove OptiMEM media and refresh HEK293T plates with 10 ml of growth media (again pipetting gently onto the side of the plate).

8.  On Day 4 (48 hr post-transfection) and 5 (72 hr post-transfection), collect virus by removing medium and filtering through a 0.45-µm filter into a 50 ml conical tube. Pool fractions from both the days. After the two collections, there is a total of 20 ml of virus. Immediately after collection/filtration (for both time points), put the virus on ice and then transfer to 4°C for short-term or −80°C for long-term storage.

9.  Infect pMEL/hTERT/CDK4(R24C)/p53DD/NRAS$^{G12D}$ melanocytes with virus to generate stable cells lines.

    A. Day 1: seed NRAS$^{G12D}$ cells at 50% confluence in 6 cm plates.

        i. Melanocytes should be maintained in Ham's F10 medium supplemented with 10% heat inactivated FBS and 1% penicillin/streptomycin at 37°C with 5% $CO_2$.

    B. Day 2: remove media and replace with 3 ml of viral supernatant containing 8 µg/ml polybrene.

        i. Incubate cells for 24 hr.

    C. Day 3: remove viral media and replace with fresh growth media.
    D. Day 4: replace growth media with fresh media containing 5 µg/ml Blastocidin.
    E. Days 4–9: select cells for ~5 days, confirming that a plate of non-transduced NRAS$^{G12D}$ cells is negatively selected in parallel.
    F. Day 9: remove Blastocidin media and expand cells into fresh growth media. Collect entire population of transduced cells for further analysis.

## Deliverables

- Data to be collected:

  1. Sequencing information and gel-verification of *PREX2* plasmids cloned into the pLenti6.3/V5 vector
  2. Mycoplasma testing of NRAS[G12D] melanocytes
  3. STR profile of NRAS[G12D] melanocytes

- Sample delivered for further analysis:

  1. NRAS[G12D] melanocytes stably expressing PREX2 mutant isoforms for further analysis (Protocols 2 and 3).

## Confirmatory analysis plan

- Statistical analysis of the Replication Data:

  1. Not applicable.

## Known differences from the original study

This replication is only generating stable melanocyte lines for GFP, wild-type PREX2, PREX2 Q1430*, and PREX2 G844D. The original study also included several other PREX2 mutants, including PREX2 K278*, E824*, P948S, and G106E. This replication will include the additional step of sequencing the endogenous *PREX2* gene in the NRAS[G12D] melanocyte cell line to determine its mutational status. All known differences in reagents and supplies are listed in the materials and reagents section above, with the originally used item listed in the comments section. All differences have the same capabilities as the original and are not expected to alter the experimental design.

## Provisions for quality control

The cell line used in this experiment will undergo STR profiling to confirm its identity and will be sent for mycoplasma testing to ensure there is no contamination. *PREX2* expression constructs obtained from the original authors will be verified for sequence identity and DNA integrity. The endogenous mutational status of *PREX2* in NRAS[G12D] melanocytes will be assessed. All data obtained from the experiment will be made publicly available, either in the published manuscript or as an open access dataset available on the Open Science Framework (https://osf.io/jvpnw/).

## Protocol 2: confirming ectopic expression of PREX2 mutant isoforms by Western blot

This protocol investigates the expression levels of mutant PREX2 isoforms in virally transduced NRAS[G12D] melanocytes that were generated in Protocol 1. This protocol uses an anti-V5 antibody to recognize tagged forms of wild-type and mutant PREX2 (as well as the GFP control), thus verifying the successful lentiviral transduction of expression constructs and providing information about ectopic protein expression levels (as was demonstrated in Figure 6A). Membranes will also be probed with anti-α-tubulin to provide normalized values of relative protein expression. Three original cell lines produced by the original authors will also be included so that protein expression levels can be compared between the two studies.

### Sampling

1. The original data presented is qualitative and this prevents power calculations being performed a priori to determine sample size (number of biological replicates). Instead, we will be including three cell lines originally derived by the authors and analyzing these cell lines in parallel to the newly derived cell lines from Protocol 1.
2. Three separate lysates will be prepared from each cell line:

   - GFP vector stable NRAS[G12D] cells (control)
   - Previously generated PREX2 Q1430* stable NRAS[G12D] cells (control from original study authors)
   - Previously generated PREX2 G844D stable NRAS[G12D] cells (control from original study authors)

- Previously generated WT PREX2 stable NRAS$^{G12D}$ cells (control from original study authors)
- PREX2 WT stable NRAS$^{G12D}$ cells (from Protocol 1)
- PREX2 Q1430* stable NRAS$^{G12D}$ cells (from Protocol 1)
- PREX2 G844D stable NRAS$^{G12D}$ cells (from Protocol 1)

1. Blots will be probed with the following antibodies:

   1. Anti-V5 tag
   2. Anti-PREX2
   3. Anti-alpha tubulin

## Materials and reagents

| Reagent | Type | Manufacturer | Catalog # | Comments |
|---|---|---|---|---|
| NRAS$^{G12D}$ melanocytes expressing GFP | Cell line | Produced in Protocol 1 | | |
| NRAS$^{G12D}$ melanocytes expressing WT PREX2 | Cell line | Produced in Protocol 1 | | |
| NRAS$^{G12D}$ melanocytes expressing PREX2 Q1430* | Cell line | Produced in Protocol 1 | | |
| NRAS$^{G12D}$ melanocytes expressing G844D | Cell line | Produced in Protocol 1 | | |
| NRAS$^{G12D}$ melanocytes expressing WT PREX2 | Cell line | Obtained from original authors | | |
| NRAS$^{G12D}$ melanocytes expressing PREX2 Q1430* | Cell line | Obtained from original authors | | |
| NRAS$^{G12D}$ melanocytes expressing G844D | Cell line | Obtained from original authors | | |
| Ham's F10 medium | Cell culture reagent | Sigma-Aldrich | N6908 | Replaces Invitrogen cat. no. 11550-043 used in original study |
| Fetal bovine serum (FBS); heat inactivated | Cell culture reagent | Sigma-Aldrich | F4135 | Replaces Invitrogen cat. no. 10082-147 used in original study |
| Penicillin–streptomycin solution (100x) stabilized | Cell culture reagent | Sigma-Aldrich | P4333 | Replaces Invitrogen cat. no. 15140-122 used in original study |
| IGEPAL CA-630 (NP-40 substitute) | Reagent | Sigma-Aldrich | I8896 | Replaces US Biological cat. no. N3500 used in original study |
| Phenylmethanesulfonyl fluoride (PMSF) | Reagent | Sigma-Aldrich | 78,830 | Replaces Pierce cat. no. 36978 used in original study |
| Protease inhibitor cocktail (mammalian) | Reagent | Sigma-Aldrich | P8340 | Replaces Roche cat. no. 11836153001 used in original study |
| Phosphatase inhibitor cocktail 2 | Reagent | Sigma-Aldrich | P5726 | Replaces Roche cat. no. 04906837001 used in original study |
| Coomassie (Bradford) Protein Assay Kit | Reagent | Thermo-Fisher (Pierce) | PI-23200 | Original brand not specified |
| BCA Protein Assay Kit | Reagent | Thermo-Fisher (Pierce) | 23227 | Original brand not specified |
| 10-cm tissue culture dishes | Labware | Corning (Sigma-Aldrich) | CLS430167 | Original brand not specified |
| Novex 4-12% Tris-Glycine, Mini, 1.0 mm, 12-well | Reagent | Life Technologies | EC60352 | |

*Table 2. Continued on next page*

*Table 2. Continued*

| Reagent | Type | Manufacturer | Catalog # | Comments |
|---|---|---|---|---|
| Novex Tris-Glycine SDS Running Buffer (10X) | Reagent | Life Technologies | LC2675 | Original brand not specified |
| Novex Tris-Glycine SDS Sample Buffer (2X) | Reagent | Life Technologies | LC2676 | Original brand not specified |
| NuPAGE® Sample Reducing Agent (10X) | Reagent | Life Technologies | NP0009 | Original brand not specified |
| ECL DualVue Western Markers (15 to 150 kDa) | Reagent | Sigma-Aldrich | GERPN810 | Original brand not specified |
| BLUEeye prestained protein ladder | Reagent | Sigma-Aldrich | 94964 | Original brand not specified |
| Nitrocellulose membrane | Reagent | BioRad | 162-0113 | Original brand not specified |
| Ponceau S solution | Reagent | Sigma-Aldrich | P7170 | Original brand not specified |
| Mouse anti-V5 tag | Antibody | Invitrogen | 451098 | |
| Mouse anti-α-tubulin, clone DM1A | Antibody | Sigma-Aldrich | T9026 | |
| Mouse anti-PREX2 | Antibody | Abcam | Ab169027 | Additional reagent not used in original study |
| Horse anti-mouse IgG, HRP-linked antibody | Antibody | Cell Signaling Technologies (CST) | 7076 | |
| Tris Buffered Saline (TBS) | Reagent | Sigma-Aldrich | T5912 | Replaces Fisher cat. no. BP2471-1 used in original study |
| Tween 20 | Reagent | Sigma-Aldrich | P1379 | Original brand not specified |
| ECL Prime Western Blotting Detection Reagent | Reagent | Sigma-Aldrich (GE Healthcare) | GERPN2236 | Replaces Pierce cat. no. 34075 used in original study |

## Procedure

1. Maintain NRAS[G12D] melanocyte lines in Ham's F10 medium with 10% heat inactivated FBS and 1% penicillin/streptomycin at 37°C with 5% $CO_2$.
2. Subculture the four cell lines onto three 10-cm plates each, for a total of 12 plates. These plates constitute replicates for each cell line for eventual quantitation of protein expression. Allow cells to grow to log phase.
3. Place 10 cm plates of log-phase growing cells on ice. Use a cell-scraper to scrape cells (on ice) into a microcentrifuge tube. Add 250 µl of lysis buffer per 10 cm plate.

    A. Lysis buffer = 20 mM Tris–HCl, pH 8.0, 150 mM NaCl, 2 mM EDTA, 1% NP40, 1 mM PMSF, 1× protease inhibitor cocktail, and 1× phosphatase inhibitor.
    B. Prepare three separate lysates for each cell line (one lysate scraped from each plate).

4. Incubate cells in lysis buffer for 20 min at RT. Centrifuge lysate for 15 min at 14,000 rpm at 4°C. Transfer supernatant to a fresh tube, then add 2× sample loading buffer containing reducing agent.
5. Load lysate onto a pre-cast polyacrylamide 4–12% Tris–glycine gel with molecular weight ladder.

    A. Quantify lysate total protein concentration.
    B. Load 30–50 µg of total protein per well.

6. Perform electrophoresis in standard Tris–glycine–SDS running buffer.
7. Transfer the gel onto a nitrocellulose membrane.

    A. Transfer buffer = 25 mM Tris–HCl, 192 mM glycine, 20% methanol.

 B. Use standard wet-transfer for 1–2 hr; PREX2 is a relatively large protein (runs about 160 kDa).

 C. Following protein transfer, stain the membrane with Ponceau-S in order to detect protein levels. Scan image of stained membrane before washing.

8. Block membrane in 5% milk in 1× TBS with 0.1% Tween-20 (TBS-T) overnight at 4°C on an orbital shaker.

9. Incubate membranes with primary antibody overnight at 4°C on an orbital shaker. Dilute primary antibodies in 5% bovine serum albumin in TBS-T containing 0.05% sodium azide.

 A. Mouse anti-V5; dilute at 1:5000.

 B. Mouse anti-PREX2; use at 1 µg/ml, according to the manufacturer's instructions.

10. Wash membranes six times for 10 min each with TBS-T at room temperature (RT).

11. Incubate membranes with secondary antibody for 40 min at RT on an orbital shaker. Dilute secondary antibody in 5% milk in TBS-T.

 A. Horse anti-mouse IgG; dilute at 1:2000

12. Wash membranes six times for 10 min each with TBS-T at RT.

13. Detect chemiluminescent signal with ECL Prime Western blotting detection reagent, according to the manufacturer's instructions.

14. Strip blots for 15 min in 0.2 M NaOH, then wash membranes six times for 10 min each with TBS-T at RT.

15. Block membranes in 5% milk in TBS-T for 1 hr at RT on an orbital shaker.

16. Repeat steps 9–13 for anti-α-tubulin primary antibody. Dilute primary antibody at 1:5000. Use same secondary antibody (at same dilution) as above.

17. Quantify density of bands and normalize against α-tubulin.

## Deliverables

- Data to be collected:

    1. Images of probed membranes (full images with ladder)
    2. Scanned images of Ponceau-stained membranes, post-transfer
    3. Densitometric analyses of normalized bands, presented in a bar graph showing standard deviation across replicates for each cell line

## Confirmatory analysis plan

- Statistical analysis of the Replication Data:

    1. Means and standard deviations will be computed across replicates for each cell line.
    2. We will perform a 2-way ANOVA (2 × 3 factorial analysis), comparing expression levels of the three PREX2 variants and the two cell-line cohorts (the originally-derived cell lines and the newly-derived cell lines from Protocol 1). This analysis will test two parameters: a) whether the original and replication values are different and b) if the three PREX2 variants are different. Because our hypothesis is that they are all the same, no individual follow-up tests are needed.

## Known differences from the original study

This replication is only analyzing protein expression from cell lines engineered to express GFP, wild-type PREX2, PREX2 Q1430*, and PREX2 G844D. The original study also included several other PREX2 mutants, including PREX2 K278*, E824*, P948S, and G106E. This replication includes an antibody probing for PREX2, so that we can better determine its endogenous expression level. Additionally, we are also testing protein expression in the original PREX2 cells lines derived by the original authors, so that we can compare expression levels between the original lines and the replication lines. All known differences in reagents and supplies are listed in the materials and reagents section above, with the originally used item listed in the comments section. All differences have the same capabilities as the original and are not expected to alter the experimental design.

## Provisions for quality control

The endogenous expression of PREX2 will be assessed in cell lines not overexpressing PREX2 variants. An image of Ponceau-stained membranes (post-transfer) will be included to verify successful protein transfer. All of the raw data, including the image files and quantified bands from the Western blot, will be uploaded to the project page on the OSF (https://osf.io/jvpnw/) and made publically available. This experiment is also the quality control for the other replication protocols as it assesses the levels of ectopic PREX2 variant expression in the utilized cell lines.

## Protocol 3: generation of tumor xenografts expressing mutated forms of PREX2

This protocol assesses the propensity of ectopically expressed PREX2 mutations to accelerate tumor formation of immortalized human melanocytes in vivo. This protocol utilizes stably transfected NRAS$^{G12D}$ human melanocyte lines that were previously generated and analyzed in Protocols 1 and 2. The melanocytic lines are transplanted into immunodeficient mice alongside control melanocytes expressing wild-type PREX2 or GFP (green fluorescent protein). Tumor growth is assessed for 16 weeks, and tumor-free survival is monitored, as depicted in Figure 3B, S6B. Further, confirmatory staining and analysis of tumor tissue will be completed, as depicted in Figure 6C.

## Sampling

- These experiments will utilize 7, 8, or 14 mice per treatment group, for a total power of ≥80%.

  1. See Power calculations section for details

- Outline of experimental conditions:

  1. NCR-NUDE female mice injected subcutaneously with:

- GFP-vector stable NRAS$^{G12D}$ melanocytes (control)

  ○ $n = 14$

- PREX2 WT stable NRAS$^{G12D}$ melanocytes (control)

  ○ $n = 7$

- PREX2 Q1430* NRAS$^{G12D}$ melanocytes

  ○ $n = 8$

- PREX2 G844D NRAS$^{G12D}$ melanocytes

  ○ $n = 14$

## Materials and reagents

| Reagent | Type | Manufacturer | Catalog # | Comments |
|---|---|---|---|---|
| NRAS$^{G12D}$ melanocytes expressing GFP | Cell line | Produced in Protocol 1 | | |
| NRAS$^{G12D}$ melanocytes expressing WT PREX2 | Cell line | Produced in Protocol 1 | | |
| NRAS$^{G12D}$ melanocytes expressing PREX2 Q1430* | Cell line | Produced in Protocol 1 | | |
| NRAS$^{G12D}$ melanocytes expressing G844D | Cell line | Produced in Protocol 1 | | |
| Ham's F10 medium | Cell culture reagent | Sigma-Aldrich | N6908 | Replaces Invitrogen cat. no. 11550-043 used in original study |

*Table 3. Continued on next page*

*Table 3. Continued*

| Reagent | Type | Manufacturer | Catalog # | Comments |
|---|---|---|---|---|
| Hanks' balanced salt solution, with sodium bicarbonate, without phenol red, calcium chloride, and magnesium sulfate | Cell culture reagent | Sigma-Aldrich | H6648 | Original brand not specified |
| Matrigel Matrix High Concentration (HC), phenol red-free | Cell culture reagent | Corning | 354262 | Original catalog number not specified |
| Fetal bovine serum (FBS); heat inactivated | Cell culture reagent | Sigma-Aldrich | F4135 | Replaces Invitrogen cat. no. 10082-147 used in original study |
| Penicillin–streptomycin solution (100x) stabilized | Cell culture reagent | Sigma-Aldrich | P4333 | Replaces Invitrogen cat. no. 15140-122 used in original study |
| 1 mL syringe; 26 G x 5/8 needle (single-use) | Labware | BD Biosciences | 309597 | Original brand not specified |
| NCR-NUDE mice(homozygous; NCRNU-F) | Mouse line | Taconic | NCRNU-F | |
| Carazzi's *Haematoxylin* | IHC Stain | Specific brand information will be left up to the discretion of the replicating lab and recorded later | | |
| Eosin | IHC Stain | | | |
| Permount | Mounting medium | | | |

## Procedure

1. Maintain NRAS$^{G12D}$ cell lines in Ham's F10 medium with 10% heat inactivated FBS and 1% penicillin/streptomycin at 37°C with 5% $CO_2$.
2. Resuspend $1 \times 10^6$ cells in a 1:1 ratio of Matrigel and Hanks' balanced salt solution and keep on ice. The final injection volume should be 100 µl.
3. Subcutaneously inject $1 \times 10^6$ cells with a 26-gauge needle and insulin syringe into 6- to 8-week old female NCR-NUDE mice.

   A. Mice were housed per IACUC regulations, in barrier housing with standard chow and 12 hr light/dark cycles.
   B. Anesthetize mice with isofluorane prior to injection.
   C. Inject mice subcutaneously on the flank.

4. Monitor mice three times a week for tumor development for 16 weeks.

   A. Record date when visible tumor is detected.

5. Measure tumor volume once weekly.

   A. Measure tumor in two directions with calipers. Calculate tumor volume as (length × width$^2$)/2.

6. Track survival of mice for 16 weeks, recording dates of euthanasia.

   A. Sacrifice mice when tumor volume reaches 1.5 cm$^3$ or if mice become moribund or cachectic.
   B. Sacrifice any surviving mice at the end of the study.

7. Upon euthanasia, harvest and process tumor tissue for further analysis.

   A. Harvest one representative tumor per mouse group (total of 4 tumors).
   B. Fix tissues in 10% neutral buffered formalin for 24 hr.
   C. Dehydrate tissues through graded alcohols and clear in xylene.
   D. Infiltrate with, and then embed, tissues in paraffin and section into 5-µm sections.
   E. Mount sections onto positively charged slides.

8. Stain tumor section with H&E (total: 1 stained section per tumor = 4 stained sections).

   A. Perform H&E staining by hand using the following procedure:

      i. Deparaffinize sections twice in xylene, then rehydrate through graded alcohols (95%, 70%, 50% ETOH) to water.
      ii. Stain sections with Carazzi's hematoxylin, then rinse slides in water.
      iii. Stain sections with eosin.
      iv. Dehydrate sections through graded alcohols (50%, 70%, 90%) and then place in xylene.
      v. Apply coverslips to slides with Permount and store slides at room temperature.

9. Blindly image stained sections and have images blindly analyzed by a Board Certified Veterinary Pathologist to verify the tumor composition of the tissue sections.

## Data to be collected

- Deliverables:

   1. Mouse health records (age, time to tumor detection, tumor incidence, date of euthanasia, and cause of termination)
   2. Raw and calculated tumor volume measurements for each date/mouse
   3. Kaplan–Meier curves generated for tumor-free survival of each mouse line
   4. Images of H&E stained tumor sections and pathology report. (compare to Figure 6C)
   5. Pathologist's report of tissue section evaluation

## Confirmatory analysis plan

This replication attempt will perform the statistical analyses listed below, compute the effects sizes, compare them against the reported effect size in the original paper, and use a meta-analytic approach to combine the original and replication effects, which will be presented as a Forest plot.

- Statistical analysis of the Replication Data:

   1. Comparison of Kaplan–Meier survival curves tracking tumor incidence using Bonferroni's correction for multiple comparisons.

      - The authors originally examined the Kaplan–Meier curves for PREX2 mutants and compared the endpoint values of the mutant curves to the endpoint values of the wild-type PREX2 curve using an unpaired two-tailed $t$-test. We will replicate their $t$-tests but also compare the entire survival curves (each mutant curve vs both wild-type and GFP control) using the log-rank Mantel–Cox test with Bonferroni's alpha correction, which we believe is a more appropriate statistical approach.

   2. Comparison of tumor growth rates

      - We will measure tumor growth rates across all mouse cohorts over the length of the study. These data were collected but not reported or analyzed in the original study. We will plot growth curves for each treatment group and use area under the curve analysis to calculate the mean and std. error. We will then use the means, std. error, and $n$ to perform a 1-way ANOVA. Further, we will perform corrected $t$-tests (Bonferroni correction) to perform pairwise comparisons between PREX2 mutants and either GFP or wild-type controls.

## Known differences from the original study

This replication is only generating and analyzing xenografts based on the stable melanocyte lines for GFP, wild-type PREX2, PREX2 Q1430*, and PREX2 G844D. The original study also generated and analyzed tumor xenografts using other PREX2 mutant-expressing melanocyte lines, including PREX2 K278*, E824*, P948S, and G106E. In order to sufficiently power all experiments and achieve the necessary number of events for Kaplan–Meier analysis, the duration of this replication will be extended from 9 weeks in the original paper to 16 weeks in the replication. All known differences in reagents and supplies are listed in the materials and reagents section above, with the originally used item listed in

the comments section. All differences have the same capabilities as the original and are not expected to alter the experimental design.

## Provisions for quality control

The genetic integrity, mycoplasma-free purity, and levels of exogenous expression of each NrasG12V melanocyte line used in this experiment have been previously validated in Protocols 1 and 2. All mice will be handled and housed in accordance with the Institutional Animal Care and Use Committee (IACUC). All data obtained from the experiment—raw data, data analysis, control data, and quality control data—will be made publicly available, either in the published manuscript or as an open access dataset available on the Open Science Framework (https://osf.io/82nfe/)

## Power calculations

### Protocol 3

### Summary of original data

**Figure 3B. Kaplan–Meier survival curves**

| Figure 3B. Kaplan–Meier survival curves | Median survival | Hazards ratio [to WT] | Hazards ratio [to GFP] | N |
|---|---|---|---|---|
| WT PREX2 | N/A | N/A | N/A | 10 |
| GFP | N/A | N/A | N/A | 10 |
| PREX2 Q1430* | 5 weeks | 0.08758 | 0.1243 | 10 |
| PREX2 G844D | 5 weeks | 0.1296 | 0.1952 | 10 |

Note: Mantel–Haenszel hazard ratios were generated in Graphpad Prism v. 6.0 following analysis of Kaplan–Meier curves with the log-rank (Mantel–Cox) test using the Mantel–Haenszel method.

### Test family

- Log-rank (Mantel–Cox) test with Bonferroni alpha correction for multiple comparisons

### Power calculations

- Performed with the Sample Size Calculator hosted by the Clinical & Translational Science Institute (CTSI) at the University of California–San Francisco (http://www.sample-size.net/sample-size-survival-analysis/) (*Rubinstein et al., 1981*; *Schoenfeld, 1983*)
- To account for multiple comparisons, a corrected alpha value of 0.0125 [0.05/4] was used in determining power calculations.

| | Experiment duration | A Priori power | Total events needed (WT or GFP) | Estimated sample size (WT or GFP) | Total events needed (PREX2 mutants) | Estimated sample size (PREX2 mutant) |
|---|---|---|---|---|---|---|
| Q1430* vs WT | 16 weeks | ≥80% | 1 | 7 | 6 | 7 |
| G844D vs WT | 16 weeks | ≥80% | 1 | 6 | 11 | 12 |
| Q1430* vs GFP | 16 weeks | ≥80% | 3 | 14 | 7 | 8 |
| G844D vs GFP | 16 weeks | ≥80% | 4 | 14 | 12 | 14 |

## Acknowledgements

The Reproducibility Project: Cancer Biology core team would like to thank the original authors, in particular Levi Garraway, Lynda Chin, and most especially Yonathan Lissanu Deribe, for generously sharing critical information as well as reagents to ensure the fidelity and quality of this replication attempt. We are grateful to Courtney Soderberg at the Center for Open Science for assistance with statistical analyses. We would also like to thank the following companies for generously donating reagents to the Reproducibility Project: Cancer Biology: American Type Culture Collection (ATCC), BioLegend, Charles River Laboratories, Corning Incorporated, DDC Medical, EMD Millipore, Harlan Laboratories, LI-COR Biosciences, Mirus Bio, Novus Biologicals, and Sigma-Aldrich.

# Additional information

## Group author details

### Reproducibility Project: Cancer Biology

Elizabeth Iorns: Science Exchange, Palo Alto, California; William Gunn: Mendeley, London, United Kingdom; Fraser Tan: Science Exchange, Palo Alto, United States; Joelle Lomax: Science Exchange, Palo Alto, United States; Timothy Errington: Center for Open Science, Charlottesville, United States

## Competing interests

DC: Noble Life Sciences is a Science Exchange associated lab. DS: BioFactura is a Science Exchange associated lab. RP:CB: EI, FT and JL are employed by, and holds shares in, Science Exchange, Inc. The other authors declare that no competing interests exist.

## Funding

| Funder | Author |
| --- | --- |
| Laura and John Arnold Foundation | Reproducibility Project: Cancer Biology |

The Reproducibility Project: Cancer Biology is funded by the Laura and John Arnold Foundation, provided to the Center for Open Science in collaboration with Science Exchange. The funder had no role in study design or the decision to submit the work for publication.

## Author contributions

DC, DS, Conception and design; AH, Drafting or revising the article; RP:CB, Conception and design, Drafting or revising the article

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
