## [Decision Letter]

Thank you for sending your work entitled “Registered report: Melanoma genome
sequencing reveals frequent PREX2 mutations” for consideration at
*eLife*. Your article has been favorably evaluated by Charles Sawyers
(Senior editor) and 4 reviewers, one of whom is a member of our Board of Reviewing
Editors.

The Reviewing editor and the other reviewers discussed their comments before we reached
this decision, and the Reviewing editor has assembled the following comments to help you
prepare a revised submission.

There are two main conclusions drawn in the Nature paper by Berger et al. First, PREX2
is identified as a frequently (14%) non-synonymously mutated gene in melanoma. Second,
that ectopic expression of cancer-associated PREX2 proteins promote melanoma genesis in
a xenograft assay in mice. This is an important study because the identification of
driver (vs passenger) mutations in cancer is critical for understanding the mechanism of
tumorigenesis.

The first conclusion is addressed by Chroscinski and colleagues in the literature
summary. The authors note that validation of this conclusion is not supported by several
studies, including meta-analysis. However, Chroscinski and colleagues state that two
papers support the conclusion that PREX2 is frequently mutated in melanoma. This is not
correct. Turajlic et al. do describe a PREX2 mutation, but the analysis is limited to a
single patient, while Furney et al. only refer to PREX2 mutations described by Berger et
al. Thus, these papers do not support the conclusion that PREX2 is frequently mutated in
melanoma. It also appears to be significant that another study of melanoma exome
sequencing by many of the same authors published 2 months after Berger et al. (Hodis et
al.) does not appear to identify PREX2 as a frequently mutated gene. Together, these
considerations do not provide support for the conclusion presented by Berger et al. that
PREX2 is frequently mutated in melanoma. This needs to be more directly addressed by
Chroscinski and colleagues: they should discuss statistical thresholds for calling
mutations significant (e.g., MutSig) and examine PREX2 mutations in public cancer genome
portals such as cBioPortal (MSK) or the Broad Tumor Portal algorithms (in addition to
reporting on the negative studies).

The second conclusion (that is addressed by re-analysis by Chroscinski and colleagues)
is important: that PREX2 is a driver mutation rather than a passenger mutation during
melanoma formation. The proposed experimental design appears to replicate the original
study. However, a number of issues were raised by the reviewers:

1) The PREX2 mutations reported by Berger et al. are present throughout the PREX2
sequence. This is more consistent with passenger mutation than driver mutation. The
restriction of the re-analysis to a limited number of mutations originally examined is
therefore problematic. To document that the PREX2 mutations acts as drivers, it would be
best to examine the same mutations that are reported in the original study, rather than
a sub-set of these mutations.

2) There are a number of problems with the original experimental design that complicate
conclusions drawn from the xenograft study. Chroscinski and colleagues should be aware
that: a) the genetic status of PREX2 in the cell line that is employed is unknown; b)
the relative expression of endogneous PREX2 and ectopically expressed PREX2 is unknown;
and c) it has previously been reported that WT PREX2 in a different cells (MCF10A)
causes PTEN inhibition, activation of AKT, and increased proliferation ([3] Science 325, 1261). These
deficiencies in the design of the original study will influence the ability to draw
sound conclusions from the study, but will be common between the original study and the
replication study.

3) Regarding the power calculations, the proposed power calculations take for granted
survival and hazard numbers published in the original study. This is fine at this stage,
however, we suggest the following improvements.

(a) Cross-study variation should be taken into account to determine expected loos of
power computed on published numbers, pre-data collection. This is hard to estimate, but
papers by Giovanni Parmigiani and collaborators at the Dana Farber provide some
estimates about cross-study variation that could be used for this purpose. The authors
should budget some additional variability because of cross-study reproducibility, and
increase the sample size on-the-fly, as they deem appropriate as deemed appropriate.

(b) The final report on the replicated study should report the actual power of the
tests, based on the standard deviations in the replicated study.

---

## [Author Response]

*There are two main conclusions drawn in the Nature paper by Berger et al. First,
PREX2 is identified as a frequently (14%) non-synonymously mutated gene in melanoma.
Second, that ectopic expression of cancer-associated PREX2 proteins promote melanoma
genesis in a xenograft assay in mice. This is an important study because the
identification of driver (vs passenger) mutations in cancer is critical for
understanding the mechanism of tumorigenesis*.

*The first conclusion is addressed by Chroscinski and colleagues in the
literature summary. The authors note that validation of this conclusion is not
supported by several studies, including meta-analysis. However, Chroscinski and
colleagues state that two papers support the conclusion that PREX2 is frequently
mutated in melanoma. This is not correct. Turajlic et al. do describe a PREX2
mutation, but the analysis is limited to a single patient, while Furney et al. only
refer to PREX2 mutations described by Berger et al. Thus, these papers do not support
the conclusion that PREX2 is frequently mutated in melanoma. It also appears to be
significant that another study of melanoma exome sequencing by many of the same
authors published 2 months after Berger et al (Hodis et al.) does not appear to
identify PREX2 as a frequently mutated gene. Together, these considerations do not
provide support for the conclusion presented by Berger et al. that PREX2 is
frequently mutated in melanoma. This needs to be more directly addressed by
Chroscinski and colleagues: they should discuss statistical thresholds for calling
mutations significant (e.g., MutSig) and examine PREX2 mutations in public cancer
genome portals such as cBioPortal (MSK) or the Broad Tumor Portal algorithms (in
addition to reporting on the negative studies)*.

We thank the reviewers for these astute suggestions. We have amended the Introduction to
more accurately describe the findings of Turajlic et al., and we have removed the
incorrectly used reference for Furney et al. We have added a sentence to highlight the
possibility of false-positive findings due to tumor heterogeneity, as described by
Lawrence, et al. We have also replaced the phrase “significantly mutated”
with “frequently mutated” in the Introduction.

In general, the Reproducibility Project: Cancer Biology focuses on generation of new
data, not reanalysis of existing datasets. As such, while interesting, we feel it is
beyond the scope of this replication study to perform data mining or analyses of
*PREX2* mutations in various genomic databases to determine prevalence
or significance. Rather, we will instead limit our focus to replicating experiments that
were performed in the original paper.

*The second conclusion (that is addressed by re-analysis by Chroscinski and
colleagues) is important: that PREX2 is a driver mutation rather than a passenger
mutation during melanoma formation. The proposed experimental design appears to
replicate the original study. However, a number of issues were raised by the
reviewers*:

*1) The PREX2 mutations reported by Berger et al. are present throughout the
PREX2 sequence. This is more consistent with passenger mutation than driver mutation.
The restriction of the re-analysis to a limited number of mutations originally
examined is therefore problematic. To document that the PREX2 mutations acts as
drivers, it would be best to examine the same mutations that are reported in the
original study, rather than a sub-set of these mutations*.

We thank the reviewers for this insightful comment. However, ultimately, the goal of the
Reproducibility Project: Cancer Biology is not to appraise the biological conclusions
and implications imparted by the original authors, but rather to systematically assess
the degree to which we can reproduce the methodology and experimental effect sizes
described in the original paper. Of note, the original authors also chose to only
analyze a subset (6/28) of all of the *PREX2* mutations they identified
(see Figure 3A).

We agree that all of the experiments included in the original study are important, and
choosing which experiments to replicate has been one of the great challenges of this
project. We acknowledge that the exclusion of certain experiments limits the scope of
what can be analyzed about the project, but we are attempting to identify a balance of
breadth of sampling for general inference with sensible investment of resources on
replication projects.

Consistent with this mentality, we feel it is beyond the scope of this replication study
to assess the original authors’ conclusions and interpretations regarding whether
*PREX2* is a passenger or driver in melanoma. We believe that
replicating a subset of the original data is appropriate to allow us achieve our goal of
quantitatively evaluating the ability of the originally reported results to be
replicated. To avoid confusion regarding this issue, we have altered the Introduction so
that it does not reference the debate over passenger versus driver mutations. We will
also refrain from discussing the functional relevance of any of the mutations that we
are not directly testing, including those identified but not analyzed in the original
paper.

*2) There are a number of problems with the original experimental design that
complicate conclusions drawn from the xenograft study. Chroscinski and colleagues
should be aware that: a) the genetic status of PREX2 in the cell line that is
employed is unknown; b) the relative expression of endogneous PREX2 and ectopically
expressed PREX2 is unknown; and c) it has previously been reported that WT PREX2 in a
different cells (MCF10A) causes PTEN inhibition, activation of AKT, and increased
proliferation (*[3]
*Science 325, 1261). These deficiencies in the design of the original study will
influence the ability to draw sound conclusions from the study, but will be common
between the original study and the replication study*.

We thank the reviewers for these suggestions. We remind the reviewers that this project
focuses on direct replication of the experiments as detailed in the original report and
with information provided by the original authors. Aspects of an experiment not included
in the original study are occasionally added to ensure the quality of the research, but
by no means is a requirement of this project; rather, it is an extension of the original
work. Adding additional aspects not included in the original study can be of scientific
interest, and can be included if it is possible to balance them with the main aim of
this project: to perform a direct replication of the original experiment(s).

As such, we agree with the reviewers that there is scientific interest in better
understanding some aspects of this biological system. To address the reviewers’
concern (a) about the genetic status of endogenous *PREX2*, we have added
an additional step to Protocol 1, whereby the endogenous *PREX2* gene in
NRAS^G12D^ melanocytes will be sequenced to determine its mutational status.
Additionally, to address the reviewers’ concern (b) about the expression levels
of endogenous PREX2, we have added an additional step to Protocol 2, whereby we will
also blot for PREX2 protein in both the overexpressed PREX2 variants and the GFP vector
control, comparing the levels of PREX2 expression across cell lines. However, we will
consider these data exploratory and therefore not include them in our statistical
analysis. As for the reviewers’ concern (c) regarding the behavior of PREX2 in
different cell lines, we believe addressing this concern is beyond the scope of this
particular replication study.

*3) Regarding the power calculations, the proposed power calculations take for
granted survival and hazard numbers published in the original study. This is fine at
this stage, however, we suggest the following improvements*.

*(a) Cross-study variation should be taken into account to determine expected
loos of power computed on published numbers, pre-data collection. This is hard to
estimate, but papers by Giovanni Parmigiani and collaborators at the Dana Farber
provide some estimates about cross-study variation that could be used for this
purpose. The authors should budget some additional variability because of cross-study
reproducibility, and increase the sample size on-the-fly, as they deem appropriate as
deemed appropriate*.

We thank the reviewers for these suggestions. The cross-study variation, such as
approaches that utilize the 95% confidence interval of the effect size, can be useful in
conducting power calculations when planning adequate sample sizes for detecting the true
population effect size, which requires a range of possible observed effect sizes.
However, the Reproducibility Project: Cancer Biology is designed to conduct replications
that have 80% power to detect the point estimate of the originally reported effect size.
While this has the limitation of being underpowered to detect smaller effects than what
is originally reported, this standardizes the approach across all studies to be designed
to detect the originally reported effect size with at least 80% power. Also, while the
minimum power guarantee is beneficial for observing a range of possible effect sizes,
the experiments in this replication, and all experiments in the project, are designed to
detect the originally reported effect size with a minimum power of 80%. Thus, performing
power calculations during or after data collection is not necessary in this replication
attempt as all studies included are already designed to meet a minimum power or are
identified beforehand as being underpowered and thus are not included in the
confirmatory analysis plan. The papers by Giovanni Parmigiani and collaborators
highlight the importance of accounting for variability that can occur across different
studies, specifically gene expression data. While it is possible for a difference in
variance between the originally reported results and the replication data, this will be
reflected in the presentation of the data and a possible reason for obtaining a
different effect size estimate.

*(b) The final report on the replicated study should report the actual power of
the tests, based on the standard deviations in the replicated study*.

As described above, we do not see the value in performing post-hoc power calculations on
the obtained data. However, we do agree that reporting the actual power of the tests to
detect the originally reported effect size estimate based on the sample size analyzed in
the replication study is important and will be reported.